# Hepatitis E Virus Infection—Immune Responses to an Underestimated Global Threat

**DOI:** 10.3390/cells10092281

**Published:** 2021-09-02

**Authors:** Paul Kupke, Jens M. Werner

**Affiliations:** Department of Surgery, University Hospital Regensburg, 93053 Regensburg, Germany; jens.werner@klinik.uni-regensburg.de

**Keywords:** hepatitis E virus, solid organ transplantation, innate lymphoid cells, natural killer cells, natural killer T cells, T cells

## Abstract

Infection with the hepatitis E virus (HEV) is one of the main ubiquitous causes for developing an acute hepatitis. Moreover, chronification plays a predominant role in immunocompromised patients such as transplant recipients with more frequent severe courses. Unfortunately, besides reduction of immunosuppression and off-label use of ribavirin or pegylated interferon alfa, there is currently no specific anti-viral treatment to prevent disease progression. So far, research on involved immune mechanisms induced by HEV is limited. It is very difficult to collect clinical samples especially from the early phase of infection since this is often asymptomatic. Nevertheless, it is certain that the outcome of HEV-infected patients correlates with the strength of the proceeding immune response. Several lymphoid cells have been identified in contributing either to disease progression or achieving sustained virologic response. In particular, a sufficient immune control by both CD4^+^ and CD8^+^ T cells is necessary to prevent chronic viral replication. Especially the mechanisms underlying fulminant courses are poorly understood. However, liver biopsies indicate the involvement of cytotoxic T cells in liver damage. In this review, we aimed to highlight different parts of the lymphoid immune response against HEV and point out questions that remain unanswered regarding this underestimated global threat.

## 1. Introduction

Worldwide, an infection with the hepatitis E virus (HEV) is one of the main causes for an acute hepatitis. While being asymptomatic in most healthy patients, the infection can lead to severe courses in immunocompromised patients such as solid organ transplant recipients with a high risk of a chronic infection [1,2]. Furthermore, especially in developing countries, a high morbidity and mortality is reported for pregnant women mainly in the third trimester caused by an increased risk for acute liver failure [3].

Historically, HEV was described for the first time in 1983 as a new non-A, non-B hepatitis virus when it was possible to detect novel virus-like particles in stool samples via immune electron microscopy [4]. The first well-documented outbreak of HEV occurred 1955 to 1956 in New Delhi, India, due to contaminated drinking water, though it was yet to be attributed to hepatitis A virus. It took until 1994 to identify HEV as the cause for this outbreak [5,6].

HEV is a single-stranded RNA virus with a size of 7.2 kb. Its particles show a diameter of 27–34 nm and the virions are nonenveloped in feces and bile while circulating in blood in a membrane-associated, quasi-enveloped configuration [7,8,9]. The genome consists of three open reading frames (ORF) encased by noncoding regions, a 5′ cap, and a poly-A tail. ORF1 encodes for a nonstructural polyprotein that is essential for the viral replication, ORF2 encodes for the viral capsid protein and ORF3 plays a role in the release of infectious virions from host cells [10,11].

In this review we aimed to outline relevant aspects regarding the versatile lymphoid immune response and point out open questions concerning a globally challenging disease.

## 2. The Global Threat–Epidemiological Aspects of HEV

By causing an estimated number of 20 million infections per year leading to 3.4 million symptomatic cases and 70,000 deaths plus 3000 stillbirths, HEV is a major burden for health systems around the world [12]. The human affecting species Orthohepevirus A in the family of Hepeviridae is divided into 8 different genotypes, in which HEV-5 and -6 are limited to wild boars and HEV-7 and -8 to dromedary and Bactrian camels. Since there is a report about a liver transplant recipient, whose consumption of camel meat and milk led to a chronic infection caused by HEV-7, humans can be infected by HEV-7 in rare cases [13,14]. Furthermore, recent studies from Hong Kong have shown that patients might also be infected by Orthohepevirus C genotype 1, an HEV species so far believed to be limited to rats, leading to hepatic and extrahepatic manifestations in these patients [15]. However, the genotypes primarily affecting humans are HEV-1 to -4 and they differ widely in geographical distribution, transmission, and disease progression (Figure 1). A recent study showed evidence that the induction of hepatic transcriptomes significantly deviates after infection with different HEV genotypes [16].

HEV-1 and -2 are generally limited to humans and usually transmitted through fecal-contaminated water, in the majority of cases this is the consequence of susceptible hygiene standards in combination with incidents affecting the drinking water supply [12,17]. Especially severe rainfalls and flooding in developing countries can lead to an abrupt increase in mostly self-limiting acute hepatitis cases but can also be life-threatening for distinct patient groups like pregnant women. Aside from epidemic cases, HEV infection can also occur sporadic in endemic regions. HEV-1 appears primarily in Asia, and HEV-2 in Mexico and Africa [2,3,7,18].

HEV-3 und -4 infect mammalian species, particularly pigs and wild boars. Humans play a role as accidental hosts, mainly affected through foodborne zoonotic transmission following the consumption of undercooked or raw pork and game meat, as well as even shellfish [19,20]. HEV-3 can be found around the globe, whereas HEV-4 is mostly limited to Southeast Asia, though it has been isolated from European pigs as well. Thus, HEV-3 and -4 cause sporadic, autochthonous cases of viral hepatitis in developing as well as in developed countries with a hyperendemic accumulation of HEV-3, such as in Southwest France [18,21,22,23]. In total, numbers of HEV cases are increasing; however, Mahrt et al. [24] and Faber et al. [25] hypothesized that the reason might be the rising awareness with enhanced and improved testing strategies. A peculiarity of HEV-3 and -4 in contrast to HEV-1 and -2 is the danger of a chronic infection in organ transplant recipients and immunocompromised patients in general [26,27,28,29]. Besides, HEV-3 shows a high risk of inducing extrahepatic neurological complications [30,31,32].

## 3. Underestimated Paths of HEV Transmission

As mentioned above, the main transmission routes for HEV infection are through contaminated drinking water for HEV-1 and -2, whereas HEV-3 and -4 are mostly transmitted zoonotically [12,17,19]. Nevertheless, numerous cases are reported that show how various transmission of HEV can occur. Especially in terms of elevated mortality in pregnancies, vertical transmission between mother and child plays an important role [3,33]. Rates of transmission during pregnancy reached 100% in a study from the United Arab Emirates [34], whereas other authors indicated lower percentages [3]. Bose et al. [35] demonstrated extrahepatic HEV replication in the human placenta. Furthermore, postpartum transmission from acute infected mothers to their infants commonly occurs, potentially through breastfeeding, since HEV has been isolated from maternal breast milk [34,36,37].

By all indications, sporadic and epidemic direct person-to-person transmission is also possible [38,39], however mostly in nosocomial settings [40,41,42,43]. Nevertheless, studies on high-risk groups did not identify evidence for sexual transmission [44,45]. Regarding the endangerment of immunocompromised patients, a major focus lies on occult transmission, especially through HEV-contaminated blood products. Transfusion-associated HEV infections in endemic and non-endemic regions have been described extensively, primarily in Europe [46,47,48] and Japan [49,50,51]. The prevalence of HEV in blood donors should not be underestimated and definitely poses a threat for vulnerable patient groups [52,53,54]. Schlosser et al. [55] described a case of a liver transplant recipient with an occult HEV infection that led subsequently to a chronic infection with liver cirrhosis.

## 4. Clinical Treatment of HEV for High-Risk Patients

With asymptomatic and self-limiting courses in most healthy individuals, HEV is nevertheless also leading to severe courses in distinct patient groups. In two of three immunocompromised solid organ transplant recipients, an acute infection leads to a persisting chronic infection, and in ten percent of the patients to a subsequent liver cirrhosis [1,56,57,58,59]. As shown in Figure 2, the first-line treatment of these patients is a reduction of the immunosuppressive medication as much as possible, which already helps to clear HEV in thirty percent of the patients [1,60,61]. However, the increased protection against infection secondarily leads to an elevated risk of organ rejection. In order to maintain the balance between infection and rejection, Torque teno virus may play an pivotal role, as it is considered an endogenous marker for monitoring immune functions in SOT patients [62].

In order to treat severe courses in chronic HEV-infected patients, who did not respond to a decrease in immunosuppression, several therapeutic agents have been tested for their off-label use against HEV. The current standard of care for patients unable to clear HEV by themselves is a three-month therapy with ribavirin. It is a nucleoside analogue that has been used in the treatment of hepatitis C virus infection [57]. Furthermore, it has been shown that ribavirin is able to decrease viral replication in chronic HEV patients and therefore to lead to a sustained virologic response in 78% of the patients. This response rate could be further increased to 85% if the treatment was continued for up to six months [61,63,64]. The main severe side effect of ribavirin is a hemolytic anemia, making it necessary to reduce or discontinue the treatment [65].

To treat patients with insufficient outcomes, alternative or combination therapies have been tested, especially drugs used effectively in other viral hepatitis infections, such as pegylated interferon alfa. It has been shown in chronic HEV-infected liver transplant recipients that a three-month treatment with pegylated interferon alfa results in HEV clearance; however, by stimulating the immune system, this treatment has severe side effects and an increased risk for acute rejection [66,67,68,69,70]. Recent attention has been focused on sofosbuvir which is part of the current first-line therapy for chronic hepatitis C infection. It has been shown in in vitro experiments that sofosbuvir inhibits HEV replication and it also had an additive effect in combination with ribavirin [71]. Furthermore, cases of chronic HEV-infected patients have been described in which adding sofosbuvir to ribavirin led to a temporary HEV eradication, highlighting sofosbuvir to be a promising agent for further studies [72,73,74,75]. Nevertheless, chronic HEV patients treated with sofosbuvir monotherapy indeed showed decreased HEV replications but did not achieve viral clearances [76]. Currently, another focus lies on T cell-based immunotherapy [77]. Initially developed for the therapy of malignant and infectious diseases, it is based on cytotoxic T cells targeting specific antigens. Soon et al. [78,79] recently identified several HEV-specific T cell receptors that could play an essential role as potential candidates in the therapy of chronic HEV infections.

## 5. Structural Limitations in HEV Research

Due to its ordinarily asymptomatic clinical course, it is largely difficult to recruit broad patient populations for studies of HEV infections. In a majority of cases, observations on HEV are based on patients with pre-existing morbidity and severe disease progressions. Thus, generating patient samples in early stages of the disease is rather difficult to accomplish.

Furthermore, various animal models have been created and effectively used, but at the moment, the ideal model to comprise all facets during infection, incubation period over clinical outbreak, eventual chronification until achieving sustained virological response in humans still needs to be established [80]. Since HEV was identified, various attempts have been made to generate sufficient cell culture models to simulate HEV infection in vitro. However, these systems showed numerous limitations concerning aspects such as reproducibility, maintenance, and sufficient viral replication [81,82]. Further restrictions apply regarding the comparability of the used cell species with the human in vivo setting, because most systems are based on cancer-derived liver or lung cell lines [83]. However, it was possible to improve the used cell culture models and adapted virus isolates [81,82] and to establish novel systems based on hepatocyte-like cells [81,83]. This allows to approach the clinical setting as far as possible. Nevertheless, fundamental limitations of cell culture systems still apply [84].

## 6. Immune Responses Induced against HEV

In order to understand the context of HEV chronification and immunosuppression, it is necessary to outline different innate and adaptive humoral or cellular immune responses triggered by an HEV infection subsequently leading to viral clearance (Figure 3).

### 6.1. Adaptive Lymphoid Cells

#### 6.1.1. Humoral Response

After proliferation and differentiation from B cells, plasma cells and memory B cells are capable to produce massive quantities of antibodies. As part of the humoral immune response, those immunoglobulins are used to identify and neutralize antigens and play an important role in clearance of viral pathogens. Moreover, antibodies are broadly identified in viral diagnostics and they provide insights concerning the progression of infection [85].

In acute HEV-infected patients, anti-HEV IgM antibodies reach their peak after 6 weeks with a delayed increase of long-lasting anti-HEV IgG [2,7]. In reports about protection against HEV reinfections, animal experiments with primates have shown a correlation between persisting high-avid IgG and reduced rates of reinfection [86,87,88], in some studies even protective immunity or cross-protection between different HEV strains [89,90,91]. Furthermore, it has been observed that in particular HEV reinfections led to courses with shorter viremia, low RNA levels, lacking IgM responses, and no detectable increase of liver enzymes [88]. A study from East China showed a decrease in the frequency and severity of HEV reinfections in patients with pre-existing immunity against HEV [92]. It is important to note that antibodies obtained by infection or vaccination do not provide sterilizing immunity [70,93]. Thus, infections with subsequent circulating anti-HEV IgG can lead to attenuated courses, but do not provide life-long protection [94,95]. While anti-HEV IgM is undetectable after 32 weeks, IgG persists for years and decades, perhaps even a lifetime. During this time, IgG decreases in a period of 5 years with increasing avidity percentages. Simultaneously, the rate of patients developing seronegativity is vanishingly low [2,96,97]. The main target of neutralizing anti-HEV immunoglobulins is the ORF2 segment. However, as a result of its exosomes-associated quasi-enveloped configuration, HEV is able to prevent recognition by antibodies [98,99].

#### 6.1.2. TCR α/β T Cell Response

In viral hepatitis, T cells play a decisive role in the development of a chronic infection over spontaneous clearance. In particular, a multi-functional CD8^+^ cytotoxic T cell response supported by CD4^+^ helper T cells is necessary to achieve a sustained virological response. Furthermore, the development of memory T cells plays a crucial role in cases of reinfection [100,101]. The adaptive T cell response in HEV infection differs widely whether the patient suffered from an acute and uncomplicated infection, a chronic course with enduring viral replication or a fulminant progress with impaired liver function and severe symptoms which can develop from both acute and chronic courses.

An acute HEV infection is associated with elevated T cell frequencies. Studies have shown that CD4^+^ [102], as well as CD8^+^ [103,104], and CD4^+^CD8^+^ [104] T cell populations increase. In this context, they are more activated [103] and produce increased quantities of IFNγ [102,104,105,106] as well as IL-10 [106]. Furthermore, Tripathy et al. [107] identified an elevated frequency and an enhanced IL-10 response of regulatory CD4^+^FoxP3^+^ T cells. Taken together, this suggests a balanced regulation by pro- and anti-inflammatory cytokines in uncomplicated HEV infection.

In chronically HEV-infected patients, decreased lymphocyte counts [108] and attenuated CD4^+^ and CD8^+^ T cell responses [109] were observed. However, these changes normalized after viral clearance [109].

Interestingly, some patients develop fulminant courses. It has been concluded that the main reason for this are host-specific factors [110]. Wu et al. [106] detected an increased frequency of CD4^+^ T cells and increased Th2 cytokines, with a concomitant decrease in IFNγ production. This was confirmed by Srivastava et al. [111], who observed a decrease in IFNγ- and TNFα-producing CD4^+^ T cells. Furthermore, Wu et al. [106] identified a correlation between outcome and IFNγ production, with no association with viral replication. It is important to note that studies of peripheral lymphoid cells do not necessarily reflect the situation ongoing in the infected liver. Unfortunately, biopsies of HEV infected patients are difficultly to achieve. Nevertheless, few studies examined post-mortem liver biopsies of patients with liver failure due to HEV infection. Thereby it was remarkable that the predominant infiltrating population were CD8^+^ T cells [112,113]. Additionally, Prabhu et al. [112] underlined the absence of CD4^+^FoxP3^+^ regulatory T cells.

### 6.2. Innate-Like Lymphoid Cells

Innate-like lymphoid cells represent a link between innate and adaptive lymphoid cells. Innate-like T cells (ILTCs) combine several properties, including the expression of a functional T cell receptor (TCR) and the surveillance of cell surfaces within tissues. In doing so, they rapidly register signs of dysregulation [114,115]. In general, their immune response covers a broad spectrum. It is dependent on various factors such as TCR integration, co-stimulation, cytokine-driven signaling, and NK cell receptor interaction [115,116]. The three main groups within ILTCs distinguished are natural killer T (NKT) cells, mucosal associated invariant T (MAIT) cells, and gamma delta (γδ) T cells. Together, they account for about 10% of all circulating T cells [116]. The proportions vary greatly depending on the environment and are particularly elevated among tissue resident T cells. MAIT and NKT cells are especially abundant in the liver and lung, whereas γδ T cells accumulate mainly in mucosal tissues [116,117,118].

NKT cells are a unique innate-like lymphoid cell population sharing both NK cell and T cell attributes. NKT cells take part in the regulation of liver immunity during viral hepatitis by direct cytotoxicity and the production of large quantities of cytokines, including mediators for enhanced neutrophil infiltration [119,120]. In acute HEV infection, patients show decreased counts of circulating NKT cells, whereas activation is markedly increased [121].

Accounting for over 45% of all lymphocytes in the liver, CD161^high^ MAIT cells are the largest subpopulation of unconventional T cells. The TCR of MAIT cells is composed of invariant TCR α chains and a repertoire of Vβ chains, mostly Vα7.2 and Jα33 combined with Vβ13.2 and Vβ2. A special aspect of MAIT cells is the recognition of vitamin B metabolites bound to monomorphic MHC-like molecules (MR1) [116,122,123,124]. MAIT cells combine diverse Th1/Th17 functions with direct granzyme- and perforin-driven cytotoxicity [125,126]. However, studies about the role of MAIT cells in the context of a HEV infection are still missing.

Another subpopulation of innate-like lymphoid cells examined in HEV infection are γδ T cells. In humans, they are functionally classified according to their expression of γ and δ TCR chains. Most commonly, Vγ9^+^Vδ2^+^ T cells are contrasted with Vδ2^−^ γδ T cells [127,128]. Usually seen as strong immune defense in tumoral and viral disease, γδ T cells are enriched in liver tissue. They participate in liver protection but also contribute to lymphocyte-mediated organ damage [129]. Acute HEV-infected solid organ transplant recipients show an activation and higher frequencies of circulating naive subsets of γδ T cells [56]. Barragué et al. [130] found a mobilization of (memory) γδ T cells in acute HEV-infected patients, presumably producing high amounts of IL-10.

### 6.3. Innate Lymphoid Cell Response

In the 1970s, lymphoid cells with the ability to recognize and eliminate virus-infected cells without prior stimulation by antigens neither cytokines were described [131,132]. Henceforth, besides those natural killer (NK) cells, several further lymphoid immune cell populations with rapid cytokine secretion mechanisms upon stimulation were identified. Following their morphological resemblance, those populations have been assembled as innate lymphoid cells (ILCs) [133].

In viral hepatitis, NK cells combine antiviral and regulatory functions, and they are regarded as an important first line of cellular immune response. In general, NK cells are regulated by the interaction of activating and inhibiting surface receptors. Inhibition is predominantly driven by the recognition of major histocompatibility complexes class I, expressed by almost every healthy cell. Upon activation, besides cell-mediated cytotoxicity, NK cells produce high amounts of IFNγ, a major cytokine in anti-viral response [133,134].

Studies with HEV-infected patients showed a diminished presence of NK cells in the peripheral blood, whereas activation was strongly increased, indicating a possible migration to affected hepatic tissue, shown by higher NK cell counts in liver biopsies. During recovery, changes in cell rates and activation normalized [112,121]. No change in NK cell mediated cytotoxicity was observed in HEV-infected patients, although the fraction of CD56^low^ predominant cytotoxic NK cells was diminished towards the accumulation of CD56^high^ mainly cytokine producing NK cells [121]. Immunohistological comparisons of severe courses of hepatitis A, B, C, and E showed the highest NK cell counts in liver biopsies from HEV infected patients [112].

Knowledge on further protagonists in cellular innate immune response to HEV infection is rare. Studies on HEV-infected Mongolian gerbils showed an enhanced activation of mast cells, a cell population largely associated with allergic reactions, but also regarded as a connection between innate and adaptive immune response [135,136]. Besides, histologic liver analyses of patients with acute HEV infection verified neutrophils as the predominant population in inflammatory cell infiltrates [137].

## 7. Clinical Link between Lymphoid Cell Impairment and HEV Outcome

As already mentioned, chronification of HEV infection is mainly seen in immunocompromised patients. It is therefore worthwhile to emphasize the reason for and the way in which these patients are immunosuppressed in order to learn more about the mechanisms involved. A recent study by Ankcorn et al. [138] investigated patients suffering from persistent HEV infection over a long period of time for underlying disorders. A history of SOT was present in approximately 60% of the patients. This was followed by patients with an underlying malignant hematological disease in 28% [138].

### 7.1. Patients after Solid Organ Transplantation

Due to a rather strong therapeutical immunosuppression to prevent rejection, SOT patients can develop several opportunistic infections. A recent meta-analysis [139] found that HEV prevalence in SOT patients is about 20%. This involved first of all liver transplant recipients. Kamar et al. [1] investigated which risk factors play a role in the development of chronic HEV infection in SOT patients. They found that the main independent factors were low platelet count and the choice of immunosuppressant. In addition, other risk factors for an increased HEV seroprevalence in SOT patients include underlying liver cirrhosis and HIV infection [140].

By Kamar et al. [1], tacrolimus in particular was found to be a risk factor, mainly because of its higher immunosuppressive effect than ciclosporin A. Both drugs play an important role since they are highly potent in targeting T cells [1]. Due to the high-affinity binding of immunophilins, a group of cytosolic protein receptors, the inhibition of the intrinsic activity of the phosphatase calcineurin is achieved. This prevents activation of the transcription factor nuclear factor of activated T cells (NFAT). Decreased NFAT activation then leads to lower IL-2 synthesis, a key cytokine in T cell activation [141,142]. Since the interplay of different T cell populations is essential for clearance of acute HEV infection, it is not surprising that over 60% of HEV-infected patients developed chronic courses in the study by Kamar et al. [1]. Investigations on chronic HEV-infected heart transplant patients have shown that, when comparing different immunosuppressive therapy regimens, only mycophenolic acid (MFA) was significantly associated with HEV clearance [143]. This finding is consistent with in vitro experiments that have demonstrated that MFA directly inhibits HEV replication, whereas calcineurin inhibitors actually increase it [65]. An antiviral effect was already presented by Pan et al. [144] in studies on hepatitis C virus infection. MFA has been shown to inhibit viral replication in vivo as well as in vitro. The exact reason for the increased HEV clearance during MFA therapy remains unclear. An essential mechanism could be the inhibition of inosine monophosphate dehydrogenase (IMPDH), which is also targeted by ribavirin [65,143,145]. By inhibiting IMPDH, MFA also effectively suppresses lymphocyte proliferation [146]. In conclusion, mainly due to the impact of immunosuppressants on T cell activation in patients with SOT, there is a certain threat for chronic HEV infection. Tacrolimus in particular represents a major risk factor. MFA, on the other hand, shows direct antiviral effects on HEV replication; thus, MFA should be considered in HEV-infected SOT patients whenever possible, although the specific effects require further investigation.

### 7.2. Patients with Hematological Diseases

Among patients with underlying hematological disease, HEV infection is mainly associated with non-Hodgkin lymphoma [147,148]. In general, HEV prevalence is higher in patients with underlying malignant hematological disease than in the general population [148]. In addition, numerous patients have been reported to suffer from severe [149,150] or prolonged [151,152,153,154] HEV disease courses. Immune insufficiency due to underlying disease or chemotherapy has been implicated as a cause, with stem cell transplantation considered a risk factor as well [147,150,155].

Due to the multitude of hematological diseases and the correspondingly versatile therapy regimens, it is important to compare which therapeutic approaches support complicated HEV infections in a more frequent pattern. It is striking that in many, sometimes severe cases, rituximab was used [147,156,157,158,159,160,161,162,163], and it was possible to intercept rising transaminases and HEV markers by reducing rituximab dosing alone [159].

Rituximab acts as a CD20 antibody and has many effects, some of which are still unknown. Via the integral membrane protein CD20, it induces apoptosis, complement-dependent and antibody-dependent cell-mediated cytotoxicity in target cells, among numerous other effects [164,165]. As a consequence, there is a highly effective depletion of CD20^+^ adult B cells, which as a main effect no longer differentiate into antibody-producing plasma cells [166]. Thus, rituximab indirectly affects T cells as well, as there is a marked reduction of CD4^+^ T cells [162,167,168,169] and to a lesser extent CD8^+^ cells [167]. Through the lack of co-stimulation by B cells, T cells show impaired differentiation [170], activation [170,171,172], and cytokine production [173]. In vitro, this effect has been confirmed [173]. Moreover, the proportion of regulatory T cells is increased under rituximab therapy [171,174]. To a small extent, rituximab also directly depletes small populations of CD20^+^ T cells and NK cells [171,174].

Since HEV infection can exacerbate the course of hematologic diseases [147], adequate therapy is essential. Frequently, infections with the genotype HEV-1 lead to fatal courses, especially if the patients are of advanced age [175,176]. Reducing immunosuppression in hematologic patients is often difficult, which is why balanced administration of ribavirin is also an important pillar in this context [147,151,163]. Von Felden et al. [147] argued that early ribavirin administration is even preferable, as reduction of immunosuppression was associated with increased mortality.

### 7.3. Further Immunocompromised Patient Groups

More than 10% of chronically infected HEV patients suffer from neither SOT nor underlying hematologic disease [138]. The remaining patients are distributed among a very heterogeneous collective.

HIV infection is, compared to the general population, disproportionately common in chronically infected HEV patients [138]. Although there is no clear correlation between HEV seroprevalence and CD4^+^ cell count in HIV patients, a CD4^+^ cell count below 200 cells/mm^3^ is a major risk factor for the development of chronic HEV courses in HIV patients [177]. It has been possible to treat acute or chronic infections by therapy with ribavirin [178,179,180], pegylated interferon alfa [181], or a combination of both [182] in several cases. Interestingly, normalization of CD4^+^ cell count alone did not result in resolution of HEV infection by itself [178].

Besides HIV-infected patients, autoimmune diseases play an important role [138]. CED patients show an increased HEV prevalence, although these could not be attributed to any specific immunosuppressive therapy [183]. This contrasts with the field of rheumatic diseases. There are numerous reports on patients receiving methotrexate-containing therapy, some of whom achieved SVR after reduction of immunosuppression [157,184,185,186]. Methotrexate has broad and multiple effects on T cells, although the exact mechanisms are not fully understood. The main modes of action under debate are primarily folate antagonism, effects on adenosine signaling, and induction of apoptosis through generating reactive oxygen species [187]. In vitro, a reduction of pro-inflammatory cytokine production has also been demonstrated [187]. Biologicals are also an important component of therapy for severe rheumatologic diseases. One frequently used group of substances are the TNFα inhibitors. Here, there are also numerous reports of complicated courses of HEV infections, which partially subsided under reduction of therapy [184,185,186,188]. This is consistent with the hypothesis that the production of pro-inflammatory cytokines such as TNFα plays an important role in the control of HEV infections.

### 7.4. Women Undergoing Alterations during Pregnancy

Pregnant women in developing countries are at increased risk from HEV infections, as the predominant genotype there, HEV-1, leads to severe courses. Ex vivo studies have shown that HEV-1 replicates more efficiently at the placenta and causes severe tissue damage compared to HEV-3 [189].

Especially during the third trimester, cases of fulminant hepatitis are frequently described, leading to mortality rates of up to 30% [3,190]. As pregnant women undergo a multifaceted change of the immune system and hormonal status, those alteration may help understanding the impact of HEV infections.

As pregnancy progresses, the adaptive immune response decreases, and the number and function of T cells and NK cells steadily decline. This indicates that the immune response is switching away from the inflammatory Th1 response. In addition, a decrease in B cells is observed [191]. At the same time, the immune barrier is strengthened, primarily by increased phagocytosis and granulocyte activity [192]. Biopsies in patients with acute liver failure showed that infiltration of CD8^+^ T cells is crucially involved in the pathogenesis [112].

The alterations of hormones also play a dominant role. Up to the third trimester, both estrogen and progesterone increase. Early in pregnancy, estrogen augments the Th1 response and thus cell-dependent immunity. Later, with higher estrogen concentrations, the Th2 response is more likely to be supported along with the humoral response [193]. Simultaneously, high concentrations of progesterone suppress the maternal immune response and affect the interaction between Th1 and Th2 cells [193]. Yang et al. [194] have shown that highly elevated levels of estrogen even promote HEV replication. They also lead to preterm delivery and fetal mortality in HEV-infected patients due to placental dysfunction [195].

Currently, the main pillar to prevent severe courses is prevention, since currently no established therapy regimens are available for pregnant women [196]. Notable drugs in the therapy of other patient groups, such as ribavirin, unfortunately carry high teratogenic potential, making them absolutely contraindicated for pregnant women. It is therefore strongly recommended that women of childbearing age use contraception if taking ribavirin themselves or having sexual contact with a person under medication [197].

## 8. Extrahepatic Manifestations Associated with HEV Infections

It is important to note that extrahepatic manifestations occur regularly after HEV infection. The pathophysiological mechanisms involved are still largely unclear [198]. Several aspects have been discussed. One is the activation of the endogenous immune defense by the viral infection, which is not limited to the primary localization, and another is the direct HEV replication in extrahepatic tissues, proven for instance in the placenta [35] as well as in the cerebrospinal fluid [32,198].

The nervous system is frequently affected. A study by Kamar et al. [32] showed that the incidence of neurological complications was 5.5%. These are mainly associated with the HEV-3 genotype [199]. According to a study from China [200], HEV-4 does not appear to contribute to neurological manifestations. Damage to the peripheral nervous system is predominant, especially Guillain–Barré syndrome and neuralgic amyotrophy, but also encephalitis or myelitis [201,202].

Following a study by Kamar et al. [203] in SOT patients, HEV infection resulted in a significant decrease in eGFR, which was normalized after HEV clearance. Biopsies showed glomerular injury and, in most cases, evidence of cryoglobulinemia, also turning negative after HEV clearance [203]. Cryoglobulin-associated glomerulonephritis is most likely triggered by the exuberant immune response to viral antigens, which has also been observed upon HBV and HCV infections [204]. Nephrological manifestations of HEV occur primarily after infection with HEV-1 and HEV-3 [205].

In addition to renal and neurological manifestations, several others have been described, primarily from the hematologic spectrum, such as thrombocytopenia and aplastic anemia [205]. Besides, acute inflammation of the pancreas occurred frequently, mostly due to infection with HEV-1 [205].

## 9. Conclusions

HEV is a pathogen leading to a huge amount of acute hepatitis cases globally. Based on the fast recruitment of innate and innate-like lymphoid cells, it is likely that they provide a potent first line of defense upon HEV infection. Furthermore, a major role is played by the adaptive T cell response in order to achieve sufficient viral control, especially concerning the development of chronic courses (Table 1). After acute infection, HEV is controlled by the interaction of CD4^+^ helper and CD8^+^ cytotoxic T cells. A special role is attributed to CD4^+^FoxP3^+^ regulatory T cells. If this machinery fails, in most cases due to immunosuppressive medication, a chronic HEV infection is more likely. Interestingly, Suneetha et al. [109] were able to retrieve the exhausted T cell response by blocking immune checkpoints such as PD-1 and CTLA-4 in vitro.

It is still uncertain why distinct patient groups tend to develop severe courses with unfavorable outcomes. It has been implied by in vitro studies that HEV is not cytopathic. Since CD8^+^ cytotoxic T cells have been identified as the predominant cell population in liver biopsies from patients with acute organ failure, it seems likely that they play a serious role in the pathogenesis of severe liver damage. The absence of regulatory T cells in these biopsies and their enhanced IL-10 response in acute HEV cases leads to the conclusion that they also take a central role in regulating the respective immune response. Furthermore, the increase of Th2-driven IL-4 production and the decrease of Th1 cytokines highlights the pivotal role of a Th2 bias in fulminant cases [106].

Nevertheless, numerous relevant questions remain unanswered: (i) Which further cell populations of the innate and innate-like immune response play a significant role? (ii) What accounts for the fact that in some cases immunocompromised patients develop severe courses eventually leading to liver failure? (iii) What is the role of cytotoxic T cells in the development of severe liver injury? Further insights in these fields may help to improve the search for new therapeutic strategies to achieve better control of complicated HEV infections in the future.

## Figures and Tables

**Figure 1 cells-10-02281-f001:**
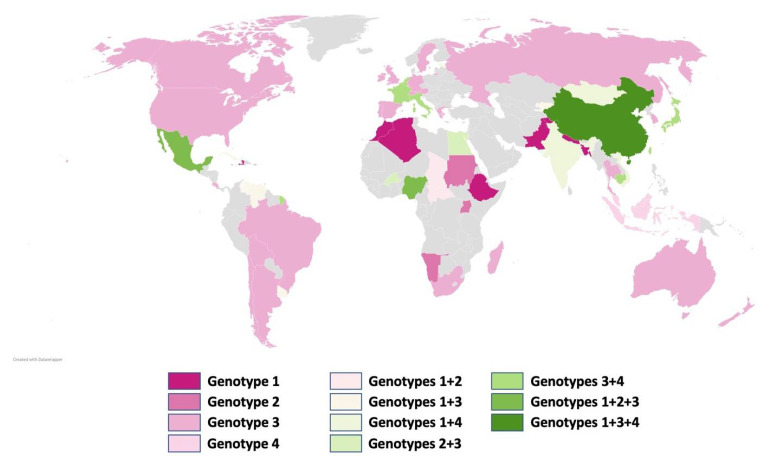
Map with the geographical distribution of the four major human pathogen HEV genotypes. Data adapted from WHO and map generated with Datawrapper, Berlin, Germany.

**Figure 2 cells-10-02281-f002:**
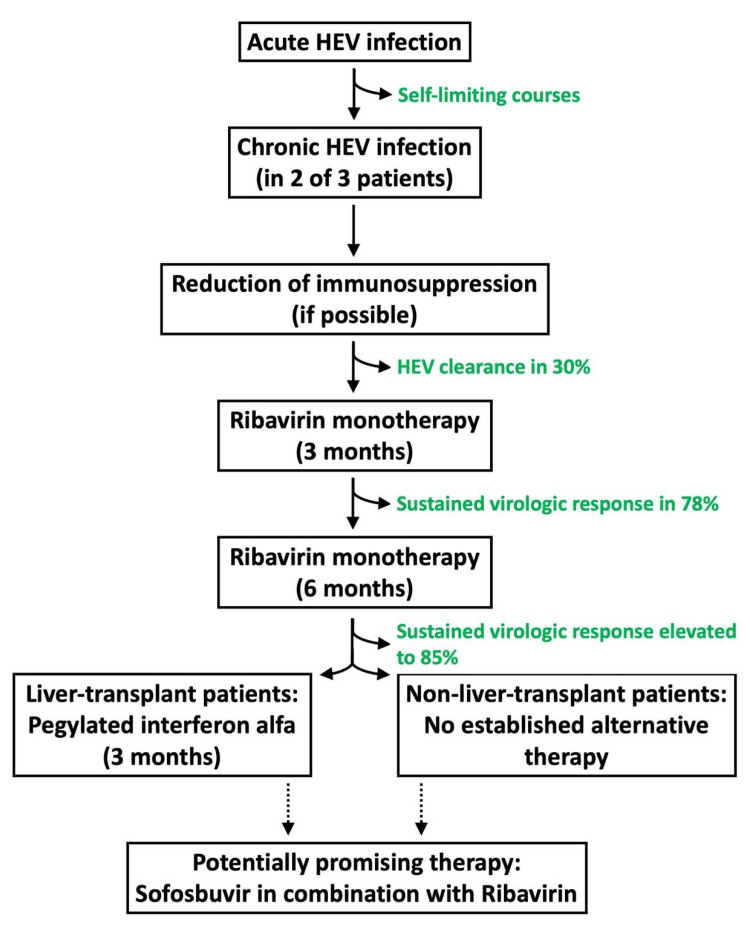
Therapy algorithm for chronic HEV-infected immunosuppressed patients.

**Figure 3 cells-10-02281-f003:**
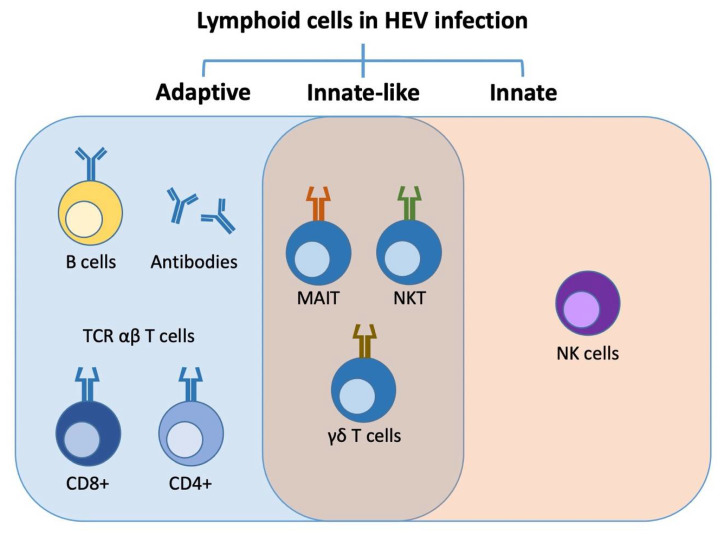
The immune response induced by HEV infection comprises the interaction of adaptive, innate-like, and innate lymphoid cells.

**Table 1 cells-10-02281-t001:** Immune response by lymphoid cells due to HEV infection.

			Lymphoid Cell Population	Immune Response	Acute HE	Chronic HE	Fulminant HE	References
**Adaptive Lymphoid Cells**			**CD4^+^ αβ T cells**	**cell count**	↑	↓	↑	[102,106,109]
				cytokine production	↑	↓		[102,106,109]
				Th1 cytokine production	↑		↓	[102,106]
				Th2 cytokine production	↑		↑	[102,106]
			CD4^+^ FoxP3^+^ Treg	cytokine production	↑			[107]
				liver infiltration			↓	[112]
			CD8^+^ ɑβ T cells	cell count	↑	↓		[103,104,105,109]
				cytokine production	↑	↓		[103,104,105,109]
				liver infiltration			↑	[112,113]
Innate-like Lymphoid Cells			NKT cells	cell count	↓			[121]
				activation	↑			[121]
			ɣδ T cells	mobilization	↑			[56,130]
				activation	↑			[56,130]
Innate Lymphoid Cells			NK cells	cell count	↓			[112,121]
				activation	↑			[112,121]
				liver infiltration			↑	[112]

## Data Availability

Not applicable.

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
