# Peer review of "Hepatitis E Virus Infection—Immune Responses to an Underestimated Global Threat"

_cells, 2021, doi:10.3390/cells10092281_

Round 1
Reviewer 1 Report
It is an interesting review presented by Kupke and Werner. The authors discussed various points: 1) The global threat – epidemiological aspects of HEV. 2)Underestimated paths of HEV transmission. 3)Clinical treatment of HEV for high-risk patients. 4)Immune responses induced against HEV. 5) Clinical link between lymphoid cell impairment and HEV outcome focusing on high-risk groups such as SOT, patients with hematological diseases, HIV-infected patients, and autoimmune diseases.
In general, the review is interesting, well written. I have a few comments/ suggestions
1) page 4 line 127: The main severe side effect of ribavirin is a
hemolytic anemia, making it necessary to reduce or discontinue the treatment
Please also consider the teratogenicity caused by the drug if given to pregnant.
2) page 11 title: Patients with hematological diseases
Please also consider that patients with hematological disorders when infected with HEV genotype 1 could lead to fatal fulminant hepatitis especially they are also old aged (PMID: 34002677, PMID: 33469320 ).
Reviewer 2 Report
HEV infection represents a major cause of concern in clinical setting, especially in immunocompromised individuals, where HEV infection tends to cronicize. The Authors provided a comprehensive overview on epidemiology and natural history, describing multiple immune pathways potentially involved in the disease onset.
Some points are reported below.
Extrahepatic manifestations of HEV are the hallmark of the systemic nature of this viral infection and likely related to immune dysfunctions (e.g. cryoglobulinemia, kidney disease, potentially neurological diseases). A major look at these features would improve the strength of the manuscript. Are there specific pathways involved?
The HIV population is underreported in this manuscript: the reduced CD4 cell population is main cause of HEV rise in HIV individuals, and it's potentially curable: these aspects should be disserted in my opinion.
Pregnancy represents a high-risk population, leading to acute liver failure and risk of miscarriage (eclampsia). Are there any insights on the immune pathways involved? More generally, a description of the study populations along all immune pathways across the studies that are described, would strengthen the translational message of the manuscript. The classical feco-oral infection (gen. 1 and 2) and zoonosis (gen 3 and 4) share the same immune dysfunctions?
It would be better to clarify that B-cell humoral response does not lead to sterilization: antibodies against HEV are not protective and this highlights one interesting point in the pathophysiological field.
